# Doubling the Accuracy of Indoor Positioning: Frequency Diversity

**DOI:** 10.3390/s20051489

**Published:** 2020-03-09

**Authors:** Berthold K.P. Horn

**Affiliations:** Department of Electrical Engineering and Computer Science, MIT, Cambridge, MA 02139, USA; bkph@csail.mit.edu

**Keywords:** indoor position, indoor location, fine time measurement, round trip time, FTM, RTT, IEEE 802.11mc, IEEE 802.11–2016, time diversity, spatial diversity, bandwidth diversity, frequency diversity, Bayesian grid, observation model, transition model

## Abstract

Determination of indoor position based on fine time measurement (FTM) of the round trip time (RTT) of a signal between an initiator (smartphone) and a responder (Wi-Fi access point) enables a number of applications. However, the accuracy currently attainable—standard deviations of 1–2 m in distance measurement under favorable circumstances—limits the range of possible applications. An emergency worker, for example, may not be able to unequivocally determine on which floor someone in need of help is in a multi-story building. The error in position depends on several factors, including the bandwidth of the RF signal, delay of the signal due to the high relative permittivity of construction materials, and the geometry-dependent “noise gain” of position determination. Errors in distance measurements have unusal properties that are exposed here. Improvements in accuracy depend on understanding all of these error sources. This paper introduces “frequency diversity,” a method for doubling the accuracy of indoor position determination using weighted averages of measurements with uncorrelated errors obtained in different channels. The properties of this method are verified experimentally with a range of responders. Finally, different ways of using the distance measurements to determine indoor position are discussed and the Bayesian grid update method shown to be more useful than others, given the non-Gaussian nature of the measurement errors.

## 1. Overview

Determining position accurately indoors, where GPS is not reliable, has many potential applications and has been of interest for some time [1,2,3,4,5,6,7,8,9,10,11,12,13,14,15] (we use the terms “position” and “location” interchangeable). One of the latest entries in this effort is fine time measurement (FTM) of round trip time (RTT) as specified in the 2016 update of the IEEE 802.11 Wi-Fi standard (also referred to as IEEE 802.11mc) reference [9,10,12,13,16].

We start by briefly discussing methods for indoor position determination. This is followed by an exploration of the error sources in indoor position determination, particularly those for FTM RTT. Then, different attempts at getting more accurate distance measurements using uncorrelated error contributions are discussed and the frequency diversity method introduced. Experimental results confirm that frequency diversity can double the accuracy of indoor position, given that there are six non-overlapping 80 MHz channels available in the 5 GHz band. Finally, various methods for determining position from distance measurements are explored and the Bayesian grid update method shown to be well suited to the task given the unusual nature of the error in distance measurement.

## 2. Introduction

The contributions of the research presented here are as follows: This paper introduces: (1) “frequency diversity”—a method for doubling the accuracy of FTM RTT distance measurements; (2) the “position-dependent error” texture surface—a new way of understanding the nature of the errors in FTM RTT distance measurement; (3) analysis of the unusual properties of the errors in distance measurement in terms of properties of super-resolution algorithms; (4) recognition of the serious impact of signal delay in common building materials resulting from their high relative permittivity—arguably more important than possible multi-path effects.

## 3. Background

A number of different methods for determining indoor position have been explored, some of which make use of properties of existing radio frequency signals emitted by Wi-Fi access points and Bluetooth beacons (For a quick review see first few chapters of [17]).

### 3.1. Received Signal Strength (RSS)

Perhaps the simplest approach is to measure the received signal strength (RSS) of a Wi-Fi access point (AP) at a hand-held device such as a smartphone (STA).

Unfortunately the inverse square law causes the accuracy to drop off inversely with distance and so the measurements are at best only useful close to the AP. Furthermore, signal strength is affected by many factors other than distance. This includes the current power level of the AP and standing waves resulting from interference between signals reflected from material outside the line of sight (LOS) between the transmitter and the receiver. It is well known that the relationship between distance and signal strength is not monotonic and not invertible (Figure 1).

### 3.2. Fingerprinting

In light of this, another way of using signal strengths has been explored. So-called “finger printing” methods depend on careful mapping signal strengths from *several* sources at points in the volume of interest. This method does not require knowledge of the positions of the APs. Signal strengths do not vary much with time as long as objects (and people) are not moved. (Some new WiFi mesh networks, such as “Linksys Velop”, include intrusion detection capability based on disturbances of signal strengths.)

When they *are* moved, the fingerprint data may have to be remeasured. Measuring signal strengths of multiple sources at many points in a volume is tedious and does not scale well.

### 3.3. Channel State Information

A simple model of the transfer function of the channel from transmitter to receiver is a weighted sum of impulses, each representing a signal that travelled along a different path. If there is a clear line of sight (LOS), the first impulse in that sum is due to the LOS path. So, if the response function can be determined, the first impulse can be isolated and used to determine the time of flight. A network analyzer can be used to measure the frequency response of a communication channel, which is the Fourier transform of the impulse response. It is, however, not practical to deploy network analyzers, in part because they require physical access to both the transmitter and the receiver.

### 3.4. Orthogonal Frequency-Division Multiplexing

In the case of orthogonal frequency-division multiplexing (OFDM) signaling—used in all but the earliest IEEE 802.11 physical layer (PHY) standards [16]—the channel is divided into many equi-spaced narrow subchannels. In operation, the response of each subchannel needs to be known and consequently is estimated continuously. This channel state information (CSI) is potentially available (at least since IEEE 802.11n using, e.g., Intel 5300).

It is a low-resolution approximation to what a network analyzer would measure. Unfortunately, at this point no widely used platform provides access to the CSI.

### 3.5. Angle of Arrival

With many antennas, a base station can estimate the direction of arrival of the signal from user equipment (smartphone) [7]. High angular resolution is required since the position error is the product of the distance and the angular resolution. Thus unless distances are very small, base stations with many antennas (and perhaps many radio chains) are needed, since angular resolution varies inversely with the number of antennas. There are also some privacy issues, since here a critical part of the position determination is done by the base stations, not the smartphone.

### 3.6. FTM RTT IEEE 802.11–2016

Finally, we come to fine time measurement (FTM) of round trip time (RTT) as specified in IEEE 802.11–2016 (also referred to as 802.11mc) [16]. One might expect this to overcome the limitations of other methods, since time of arrival is based on the *first* signal component, and so should be immune to multi-path problems, such as interference and standing waves.

Access to FTM RTT measurements has been provided on the Android platform since 2018 (Android Pie), although initially few smartphones and Wi-Fi APs supported the protocol (see also Appendix A and Appendix B).

Experimentally, one finds that the distance measurements provided by FTM RTT may have standard deviations of 1–2 m under favorable circumstances. This is fine for some applications but not others. It is important to understand the underlying causes of the observed errors in distance.

## 4. Nature of the Error

In FTM RTT, the error—difference between measurement and the actual distance—can be thought of as having several components, which behave very differently. It is important to understand these contributions to the overall error *e*, since they need to be dealt with in different ways.
(1)e=m(c;…)+E(r,c;…)+o(c;…)

Here m(c;…) is “measurement noise” (see below) which depends on the channel *c* (i.e., frequency) and other factors, while E(r;c…) is the “position-dependent error” (see below) which depends on position r, the channel *c* and other factors, while o(c;…) is the offset (see below) which depends on the channel *c*, type of initiator, type of responder etc.

All of the above also depend on the bandwidth, but, except where noted below, we’ll assume use of the highest bandwidth at which FTM RTT is supported by both the initiator and the responder (currently 80 MHz) because that normally leads to the highest accuracy.

Further, where there is a dependence on position as indicated above, there is also a dependence on orientation, which we will not continue to refer to explicitly from here on.

### 4.1. “Measurement Error”

Remarkably small spreads in results are observed when measurements are repeated without changes in position (or orientation) of initiator and responder, in a fixed environment as shown in Figure 2.

In this case, the standard deviation (e.g., 0.1–0.2 m under favorable circumstances) is considerably smaller than the actual error in distance measurement (which is typically greater than 1–2 m). As a consequence, perhaps surprisingly, results are *not* significantly improved by averaging repeated measurements.

While this error component looks a lot like typical measurement error from additive random noise, it should be noted that: (i) its distribution is *not* Gaussian; (ii) there are distant outliers in many cases (see in particular the magenta, green and cyan traces); and (iii) the distribution is often not even unimodal (see, e.g., the darkgreen trace, between 4 and 5 h). Importantly, small changes in position (or orientation) can cause large changes in the distribution. As a result repeated measurement in fixed positions can lead one to grossly underestimate the error in distance. We’ll say that repeated measurements obtained in fixed positions exploit “time diversity,” and note that time diversity does *not* provide a path to improved accuracy.

### 4.2. “Position-Dependent Error”

Perhaps somewhat surprisingly, small movements (millimeters) of the initiator (or the responder) induce large changes (meters) in reported distance measurements. This error component is a function of 3-D position (and orientation). It is difficult to explore and visualize the error dependence fully in 3-D, but much can be learned by simply scanning along lines.

It is clear that the “position-dependent” error in Figure 3 is much larger than the “measurement noise” in Figure 2. Careful measurements along lines surprisingly shows fluctuations in the error surface that have “texture element” size comparable to the wavelength of the radio frequency signal (which ranges from 58 mm for 5210 MHz to 52 mm at 5775 MHz). This is confirmed by inspection of the spatial power spectrum, which has much of its energy at and below the frequency corresponding to about two cycles per wavelength.

Measurements taken at positions separated by more than say a wavelength are fairly uncorrelated. This suggests one way of improving accuracy: average several measurements taken (far enough apart) along points spaced out along a line (or on a regular grid). This indeed leads to a result with considerably higher accuracy than averaging repeated measurements taken in a fixed position. We’ll say that repeated measurements obtained on a line (or on a grid) of positions exploit “spatial diversity” and note that spatial diversity *can* improve accuracy significantly.

It is, however, not clear how this observation can be used in practice since it requires either a set of regularly spaced antennas in an array larger than the typical smartphone, or perhaps some mechanism for moving a single antenna into a set of positions in some regular pattern.

For experiments requiring high accuracy, however, such as measurements of the relative permittivities of building materials like concrete, brick and wood, the extra effort in making measurements in several positions is well justified, since for these measurements the raw accuracy of FTM RTT is not adequate.

### 4.3. Offset

Over a large range of distances, with a clear line of site, the reported distance varies linearly with the actual distance. The slope of the linear fit is 1 (see, e.g., Figure 4) but there typically is a significant offset, which depends on the type of initiator, the type of responder, the channel in use, bandwidth, and the preamble.

Ideally, all initiator/responder combinations would come calibrated to yield zero offset. Presently different responders will yield different offsets with different initiators (sometimes differing by five or more meters). Even a particular combination of initiator and responder has different offsets when operating in different channels (which can lead to hard-to-track errors when the AP decides to switch channels for some reason!). Presently one must calibrate for the particular combination of initiator and responders to be used in order to eliminate these offsets.

### 4.4. Noise Gain

The accuracy of the final position estimate is not the same as the accuracy of the raw measurement of distance between the initiator and the responder. The error in position may be considerably larger than the error in distance measurement, depending on the geometry of the layout of responders and initiator. The ratio of the error in position to the error in the distance measurement is the “noise gain”—euphemistically referred to as “dilution of precision” (DOP) in GPS terminology [18]. This suggests that there is some benefit to carefully planning the distribution of responders so as to minimize the error in the worst-case position of the initiator (see also Appendix C and Appendix D).

### 4.5. Dependence on Bandwidth

The expected accuracy is inversely proportional to the bandwidth of the Wi-Fi signal. Currently the highest bandwidth of initiators and responders that support the IEEE 802.11 FTM RTT protocol is 80 MHz (there are some access points and some Wi-Fi adapters that support 160 MHz, but, as of this writing, do not support FTM RTT).

One may consider “bandwidth diversity” as another possible measure to improve accuracy, but the results at 40 MHz and 20 MHz tend to be noticeably worse than those at 80 MHz. As a result, there is only a small gain in accuracy using a best fit weighted average of the three results (aside from that, the offsets are different for different bandwidths and need to be calibrated out).

## 5. Where Does the Large Position-Dependent Error Come From?

The main component of the error is the position-dependent error. Given the size of the “texture element” of this type of error, it appears to be related to some sort of interference pattern resulting from reflections off objects that are not in the line of sight. This is quite unexpected since the first arriving component of the signal should *not* be affected by reflections.

In contrast to this, signal strength (RSS), being a steady state measurement, *is* subject to large fluctuations (“fast fading”) over relatively small distances due to just such interference. (see upper plot in Figure 5).

Again, at least with a clear line of sight, the first arrival should *not* be affected by later arriving signals reflected from objects in the environment. Thus, it comes as a surprise that FTM RTT distance measurements seem to be affected by some sort of interference patterns or stationary waves (see lower plot in Figure 5). To understand how this can be, we must know more about how these measurements are made.

### Super Resolution

With OFDM modulation, demodulation is done by inverse Fourier transform of samples of the signal. For 80 MHz bandwidth, these samples are taken at 80 Msps (actually, both Q (real part) and I (imaginary part) are sampled at that rate, but that does not affect the argument here). That means that samples are taken every 12.5 ns, which corresponds to 3.75 m round trip travel of the radio-frequency (RF) wave. So, if first arrival was based merely on which sample exhibits the first sign of a rising waveform, then the (one-way) resolution would be 1.875 m. The measurement actually provided to the user has much finer resolution (RTT, for example, may be given in units of 0.1 nsec, very much smaller than the 12.5 ns sampling interval). Super-resolution methods are used to “interpolate” between known samples of the signal.

Several super-resolution methods are used, such as MUSIC, ESPRIT, and pencil matrix [19,20,21,22,23,24,25,26]. These are based on specific assumptions about the transfer function of the communication channel. In particular, it is assumed that the impulse response of the channel is a weighted sum of shifted impulses, corresponding to different components of a multi-path signal.

While the aim is to provide the user with finer resolution, such methods also have limitations. They are highly non-linear and can exhibit discontinuities and non-monotonicity. Further, information on what actual algorithms are used in the Wi-Fi initiator and in the Wi-Fi access points is not available to the user.

To illustrate the potential problem, consider first an oversimplification. A simple algorithm has arrival time estimated based on when a sample of the signal amplitude exceeds some threshold. However, one cannot use a fixed threshold for deciding when the signal arrives, since the signal can vary over several orders of magnitude (e.g., −100 dBm to −40 dBm—i.e., a ratio of a million to one in power) The threshold to determine whether the “toe” of a signal has arrived must be scaled based on the strength of the signal. But that “signal strength” can only be ascertained *later* when it has reached a peak. While the “toe” is not affected by multi-path reflections, the amplitude used for normalization *is* subject to the interference pattern. So even though the first arrival is not contaminated by interference, the threshold against which it is compared *is*. This sort of effect can give rise to the wildly fluctuating position-dependent error surface described above (see lower plot in Figure 5).

The gain of the radio is adjusted in discrete steps based on the signal strength (actually, often the reported signal strength value is derived from the current AGC setting). An effect similar to the one described above (although much larger) have been ascribed to changes in AGC settings [4].

## 6. Frequency Diversity—Six Channels

Since the position-dependent error surface has “texture” the order of the wavelength of the radio frequency signal, it stands to reason that operating at different frequencies would produce different position-dependent errors. There are six non-overlapping 80 MHz channels in the 5 GHz band (42, 58, 106, 122, 138, and 155 with center frequencies 5210, 5290, 5530, 5610, 5690, 5775 MHz). This provides for up to six measurements with uncorrelated error contributions, potentially leading to a multiplication of the error by $1/\sqrt{6}\approx 0.408$. (Note that there may be some restrictions on some channels in some parts of the world. The highest channel, for example, is not available in Japan, Israel, Turkey and South Africa).

Figure 6 shows plots of distance measurements in six channels as a function of actual position. The channels have center frequencies 5210 MHz (magenta), 5290 MHz (red), 5530 MHz (brown), 5610 MHz (green), 5690 MHz (cyan), and 5775 MHz (blue). The correlation matrix (Equation (Equation 2)) shows that the position-dependent errors in the different channels are essentially uncorrelated.
(2)$$\left[\begin{array}{cccccc} 1 & -0.05 & -0.19 & 0.21 & -0.03 & 0.12\\ -0.05 & 1 & 0.13 & -0.04 & -0.14 & -0.09\\ -0.19 & 0.13 & 1 & -0.19 & -0.09 & 0.00\\ 0.21 & -0.04 & -0.19 & 1 & -0.07 & -0.11\\ -0.03 & -0.14 & -0.09 & -0.07 & 1 & -0.04\\ 0.12 & -0.09 & 0.00 & -0.11 & -0.04 & 1 \end{array}\right]$$

The st. dev. of position-dependent errors from the six channels are 0.710 m, 0.578 m, 0.540 m, 1.163 m, 0.944 m, and 0.909 m (average 0.808 m). A simple average of the six distances has st. dev. 0.309 m, which is significantly better than the best channel on its own, and more than twice as accurate as the average.

A weighted sum—rather than a plain average—can do even better. In this particular case, with weights 0.155, 0.234, 0.301, 0.080, 0.130, and 0.100, the st. dev. comes to 0.264 m (which is only about a third of the average st. dev. of the six channels).

(The weights are obtained by simple least squares fitting to produce the smallest sum of squares of errors.) With six channels, the added refinement of least-squares weighting—as opposed to simple averaging—may not always be worth the extra effort since the relative quality of the different channels depends on the environment and will be different in different situations.

By the way, averaging FTM RTT measurements from six 80 MHz channels does *not* produce the same results as if one were to perform a single FTM RTT measurement in a channel of 480 MHz bandwidth. In the case of a single ultra-wide channel, the error would be multiplied by 1/6, not $1/\sqrt{6}$.

## 7. Frequency Diversity—Three Channels

It may not always be practical or convenient to use all six 80 MHz channels for FTM RTT distance measurements. In some situations a smaller number may be more easily accessible. Several “tri-band” mesh Wi-Fi APs (e.g., Eero Pro, Netgear Orbi and Linksys Velop) have two radios which make it easy to get measurements for at least two channels in the 5 GHz band (e.g., 5210 MHz in U-NII-1 and 5775 MHz in U-NII-3). Often also, one of the radio chains is shared between the 2.4 GHz and 5 GHz bands and if the device happens to respond to FTM RTT requests in both bands (e.g., Linksys Velop) then this opens up the possibility of taking three measurement with uncorrelated error contributions.

Taking a simple average potentially multiplies the error by $1/\sqrt{3}\approx 0.577\ldots$ (assuming similar error distributions for the three channels and with uncorrelated noise). Not as good as with six channels, but still a useful improvement. Actually, this may be a bit optimistic, since the 2.4 GHz channel is not as good as the other two, one the other hand suitable weighting of the three contributions can get one close to the ideal.

In Figure 7, the bottom three plots are for channels with center frequency (i) 5210 MHz (red), (ii) 5775 MHz (green), and (iii) 2442 MHz (blue). The correlation matrix (Equation (Equation 3)) shows that the position-dependent errors in the different channels are, once again, uncorrelated.
(3)$$\left[\begin{array}{ccc} 1 & -0.00 & 0.03\\ -0.00 & 1 & -0.13\\ 0.03 & -0.13 & 1 \end{array}\right]$$

The top plot (black) in Figure 7 is for a weighted average (weights 0.48, 0.35, and 0.17 respectively). The st. dev. of the position-dependent error in the lower three plots are 0.382 m, 0.480 m, and 0.721 m, for an average st. dev. of 0.528 m. The st. dev. of a simple average is 0.302 m (which is better than any of the individual channel st. dev.). and the st. dev. of the weighted average is 0.270 m (which is almost twice as accurate as the average channel).

Typically different chips are used for the two radio chains. In the case of Eero Pro, for example, the first 5 GHz radio (and the 2.4 GHz radio) uses the Qualcomm IPQ4019 chip, while the second 5 Ghz radio uses the Qualcomm QCA9886 SoC. These have somewhat different measurement qualities and thus weighting their contributions differently (as above) helps improve the overall result.

Finally, if three channels are not available, using two channels can already bring some improvement in accuracy relative to relying on a single channel.

## 8. High Relative Permittivity of Common Building Materials

Inside buildings, signals often have to travel through walls and floors of concrete, wood, brick, drywall or glass. These materials have high relative permittivity which slows down the signal significantly. Careful measurement of thick layers of various materials show relative permittivities, in the 8–10 range for wood, and 5–7 range for concrete, depending on moisture content and composition (the signal also is attenuated significantly, but this does not affect the time-of-arrival directly) [27,28,29,30]. Time-of-flight times the speed of light is the equivalent distance travelled in vacuum—which may be considerably larger than the actual distance. A 0.5 m thick concrete wall can, for example, add 3 or 4 m to the reported FTM RTT “distance.” This needs to be taken into account somehow in the estimation of position from distance measurements. The effect of thick walls and floors should also be a concern when planning the placements of responders.

Arguably, the effect of high relative permitivitties of building materials on distance measurements is more important than that of multi-path. Particularly reminding ourselves again that the time of first arrival should *not* be affected by reflections.

Figure 8 shows how building materials affect measurements in a three-story wooden house. Figure 9 show how building materials affect measurements in a large open plan office building. The effect there is less extreme, although over long enough distances just as significant.

## 9. Recovering Position from Distance Measurements

Once we have estimated distances from a number of AP responders in known positions, we can try and determine where the initiator is.

### 9.1. Multi-Lateration

If we are dealing with a single level building, we can treat this problem in 2-D. In this case, each measurement confines the possible position of the initiator to points on a circle with an AP at the center—or a circular annulus if we take into account uncertainty in the measurement. Two measurements lead to the intersection of two circles, which typically is two points (these two points lie on the “radical line,” which is perpendicular to the line connecting the centers of the circles). A third measurement can disambiguate if needed. Three or more measurements are typically inconsistent but can be used in a least squares fashion to reduce the error in position estimation.

This is quite analogous to finding a cellular base station from multiple LTE Timing Advance (TA) measurements—just with much finer resolution [31].

In the more general full 3-D case, each measurement confines the position of the initiator to points on the surface of a sphere with an AP at its center—or a spherical shell if we take into account uncertainty in the measurements. Two measurements restrict the solution to the intersection of two spheres, which typically is a circle (this circle lies in the “radical plane,” which is perpendicular to the line connecting the centers of the spheres). A third measurement reduces the possibilities to the intersection of a circle and a sphere, which typically occurs in two places. A fourth measurement can disambiguate if needed. Four or more measurements are typically inconsistent but can be used in a least squares fashion to reduce the error in position estimation.

### 9.2. Linear Multi-Lateration?

The equations for the circles—or spheres—are second order. They do all have the same higher order terms. Thus it is tempting to subtract them pairwise to obtain linear equations, since linear equations are easy to solve. This is a mistake. While the resulting equations yield the correct solution if the measurements are perfect, the “noise gain” is very high. That is, small errors in distance measurements translate into large errors in position. One way to understand why this happens is that we are throwing away some of the constraint provided by the measurements. For convenience of calculation, we consider the solution to be confined to the planes containing the circles of intersection, *not* to the actual circles, which is a much tighter constraint. (For mathematical details of the argument see [32]). (By the way, distances to more APs are needed when using the “linearized” method than when using the full constraint).

An aside: this is quite analogous to the infamous “8-point method” in machine vision for solving the relative orientation problem. While it is very appealing because of the linear form of the equations, minimization of errors in those equations does not minimize the sum of errors in image positions [33,34]. As a result, this method cannot be recommended (other than perhaps in the hope of finding plausible starting values for methods that do the right thing).

### 9.3. Least Squares Minimization and Brute Force Search

For a given hypothesized position for the initiator, the distance from each AP can be computed and compared with the measured distance. One can then find the position that minimizes the sum of squares of the differences between computed and measured distances. Gradient-descent may not work reliably to find the global minimum of this error sum, since the shape of the error surface can be complex. We can, however, divide the space into pixels (2-D) or voxels (3-D) and simply compute the error for each cell. This is not computationally expensive, since, given the limited accuracy of FTM RTT measurements, the cells need not be very small (e.g., perhaps 0.5 m on a side). So even a typical building with side lengths of tens of meters would be represented by just a few thousand cells.

### 9.4. Kalman Filtering

Kalman filtering provides a way to update an estimate of the position and an estimate of the covariance matrix of uncertainty in the estimated position ever time a measurement is made. It is based on assumptions of Gaussian noise independent of the measurement, Gaussian transition probabilities and linearity.

Unfortunately the measurement error is not Gaussian nor is it independent of the measurement itself. Further, when near one of the responders, the area of likely positions is shaped more like a kidney (i.e., part of a circular arc)—or even bimodal—rather than something that can be well approximated by a linearly stretched out Gaussian distribution.

As a consequence, Kalman filtering does not provide the best way to use the available information.

### 9.5. Particle Filter

If a probability distribution is not easily modeled in some parameterized way (such as a multi-dimensional Gaussian), then other means may be used to represent it. One such method is that of particle filters which uses weighted samples to represent a distribution. The distribution is in effect approximated by the sum of weighted impulses. At each step, the position of the particles is updated based on a transition model. The weights of the particles are adjusted based on the measurements. Particles with low weight are then discarded, while new particles are sampled to keep the overall number of particles at a desired value. Particle filers have been applied to this problem as well [35].

### 9.6. Bayesian Grid Update

Another way of dealing with a probability distribution that cannot be easily parameterized is to represent it with values on a regular grid. Sequential Bayesian updates can be applied to such a grid of probabilities. This method starts with a prior distribution (perhaps uniform). A transition model is invoked at each step which modifies the distribution based on likely movement of the initiator (e.g., a random walk). If a floor plan is available, impenetrable walls can be taken into account in the transition model. This is followed by Bayesian update based on distance measurements, which uses an observation model which estimates the probability of seeing a measurement given the actual geometric distance between a voxel and the responder.

If a single position is required as output, rather than a distribution, one can use the mode (maximum likelihood) or the centroid (expected value) of the distribution.

As with other forms of “filtering,” there can be a lag in the response when the initiator moves more rapidly than the transition model expects. Also, a bad solution may get “trapped” behind walls, when a floor plan is used to prevent “tunneling” through walls in the transition model.

### 9.7. Observation Model

Figure 10 shows a section of a sample observation model. It shows the probability of various measured “distances” on the horizontal axis in meters) given that the actual distance between initiator and responder is 10 m (i.e., the vertical red line). The actual distance is a lower bound on the measurement. It can be considerably larger since the signal may pass through building materials with large relative permittivity. In the figure, the observation model (green curve) is a piece-wise linear fit to experimental data from a three-level residence (grey histogram).

The observation model is used to update the probability at each grid cell. For each cell on the grid, the distance from the AP is known and so the appropriate slice of the observation model can be accessed. The observed FTM RTT distance is then used to look up the probability that this observation would occur, given the known actual distance for this grid point. This value is then used to multiply the current value in that cell. Optionally, the resulting grid of values can then be normalized so it once again adds up to one.

The observation model is based on measurements in the environment in which the method is to be used. While it may be different in different situations, the general nature is that the observations are biased to be larger than the actual distance and that this bias increases with distance. It does not matter whether this is primarily because of signal delay in materials of high permittivity or reflections from strong reflectors outside the direct line of sight, since both increase the reported distance. The results of the Bayesian updates seem not to be affected strongly by details of the observation model, perhaps because they are updated again when the next observation comes along. So there seeems to be little need for an observation model to fit a specific situation with great accuracy.

### 9.8. Transition Model

We use a simple transition model of a random walk of a step size based on comfortable walking speed of 1.4 m/s. In a simple implementation, this just “pushes” probabilities into neighboring cells (exept for cells on the edge of the grid). If more information is available from inertial measurement (IMU) and magnetic compass, then this can be used to refine the transition model. But the simple model appears to be adequate for position determination. A floor plan can be used to limit “forbidden transitions” such as walking through a wall. This can further improve the tracking of a position solution as the user progresses through the environment.

Figure 11 shows probability distributions on grids with cells 0.5 m on a side. The green dots mark the positions of the responders (in this case, the floor plan was not utilized to limit the transition model). For an MP4 movie showing the Bayesian grid evolve as someone moves on one level, see [36]. For an MP4 movie showing the Bayesian grid evolve as someone moves through a three-story building, see [37].

## 10. Noise Gain (a.k.a. Dilution of Precision—DOP)

The geometric arrangement of responders determines the “dilution of precision” (DOP, or “noise gain”), that one can expect in various parts of the volume of interest.

On the left in Figure 12, is shown the annulus within which the initiator position is constrained when a single, noisy distance measurement is available. In the middle is the situation when two measurements are available from responders that are more or less at right angles in directions as seen from the initiator. Plausible solutions in this favorable case are confined to a small area. On the right is the less fortunate situation where the directions to the responders are similar, and not much new information is provided by the second measurement. Correspondngly, the likely position of the initiator is not as well confined.

When close to one of the responders, the geometry becomes more intricate, and, counter-intuitively, the solution may be less well determined. This is illustrated in Figure 13.

It is generally not a good idea to have the responders close together, since then the distance measurements will be correlated and redundant. The effect of errors typically is not isotropic, but is stronger in some directions than others (as in the case of GPS, where the vertical DOP is considerably larger than the horizontal DOP, as a result of the fact that the “visible” satellites are not distributed evenly over a sphere of possible directions) [18]. In some cases curves of constant error may be quite elongated, meaning that while the position may be well defined in some directions, it is not in others. Finding the “best” layout of responders in a given 3-D volume is an open research problem.

For additional detail see Appendix D.

## 11. Conclusions

The accuracy of FTM RTT distance determination can be doubled using frequency diversity. The error in FTM RTT distance has peculiar properties (for a start, it is non-Gaussian) that derive in part from the super-resolution algorithms used. Common building materials can introduce large errors in FTM RTT distance estimates because of their high relative permittivity. Bayesian grid estimation is well suited to the task of recovering position from distance measurements given the unusual nature of the errors. The “noise gain” in position determination can be kept low by carefully planning the geometric arrangement of the access points.

## Figures and Tables

**Figure 1 sensors-20-01489-f001:**
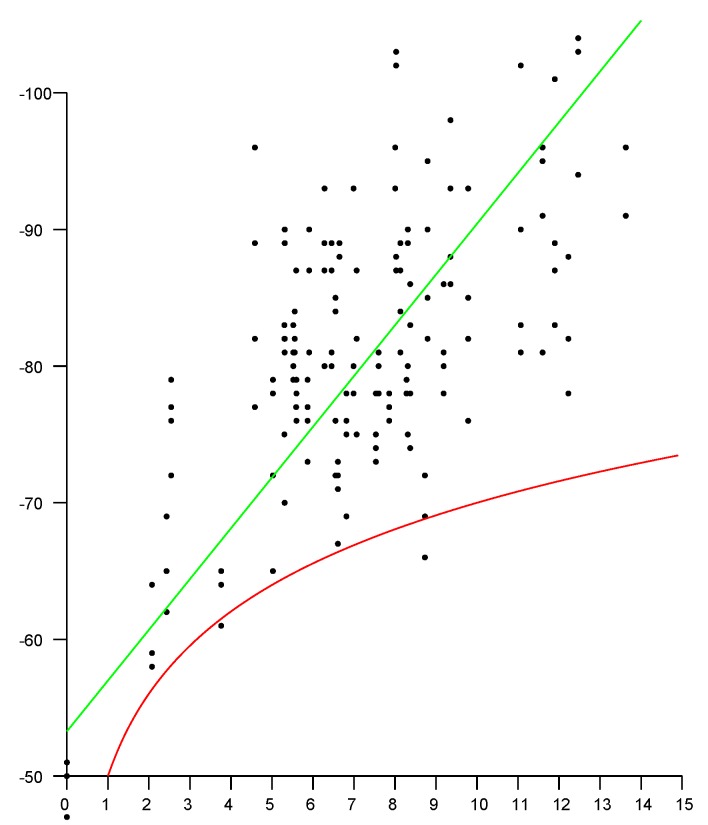
Scattergram of Signal Strength (RSS) versus distance in typical three-level wooden building. Horizontal axis: distance between smartphone and Wi-Fi access points (APs) (in meters). Vertical axis: Signal Strength (in dBm). Red curve: expected inverse square law dependence (−50−20log(R) dBm). Green line: linear fit (−53−3.7R dBm). Because of the large scatter, it should be clear that RSS is not very useful for estimating distance.

**Figure 2 sensors-20-01489-f002:**
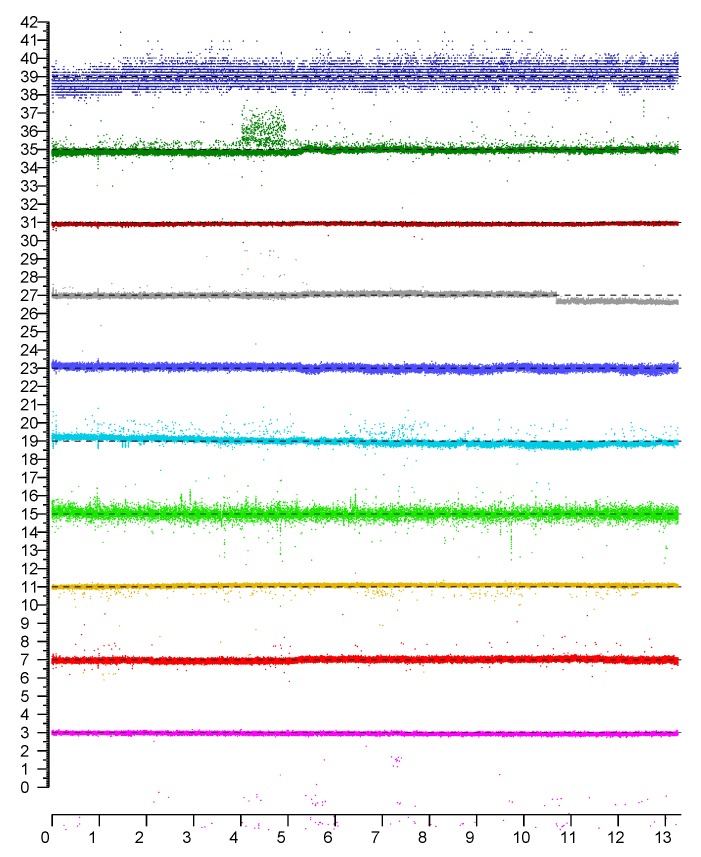
Sample measurements using ten different responders in fixed positions. (Top plot is for an access point operating in the 2.4 GHz band, the rest are in the 5 GHz band). Horizontal axis: time in hours. Vertical axis: distances in meters (individual plots are offset vertically to avoid overlap).

**Figure 3 sensors-20-01489-f003:**
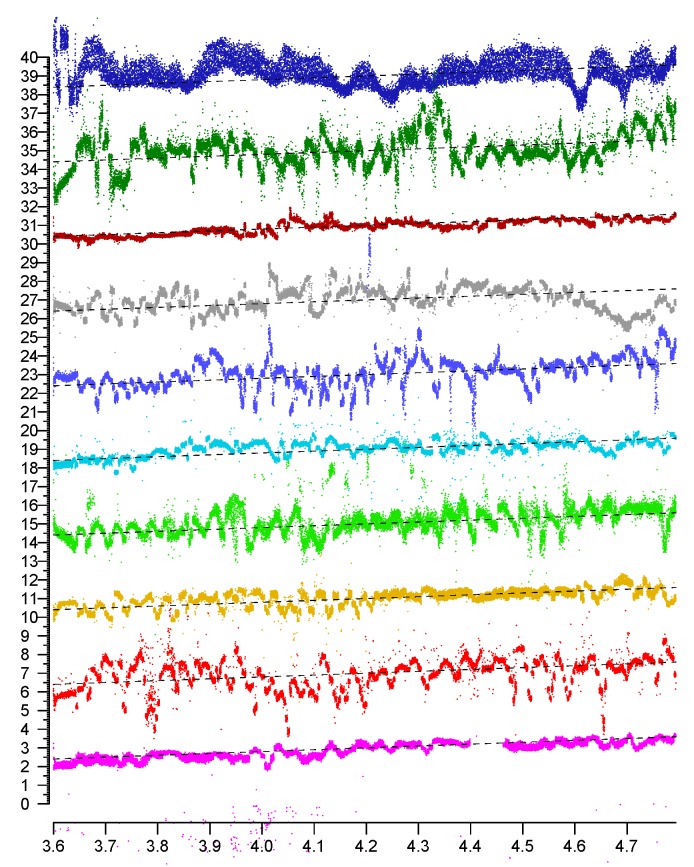
Sample measurements using ten responders in a range of positions (Top plot is for an access point operating in the 2.4 GHz band, the rest are in the 5 GHz band). Horizontal axis: actual position in meters. Vertical axis: reported distances in meters. (Note that the average slope of the plots is not equal to one, because the scales on the vertical axis and horizontal axes are different). (Individual plots are offset vertically to avoid overlap).

**Figure 4 sensors-20-01489-f004:**
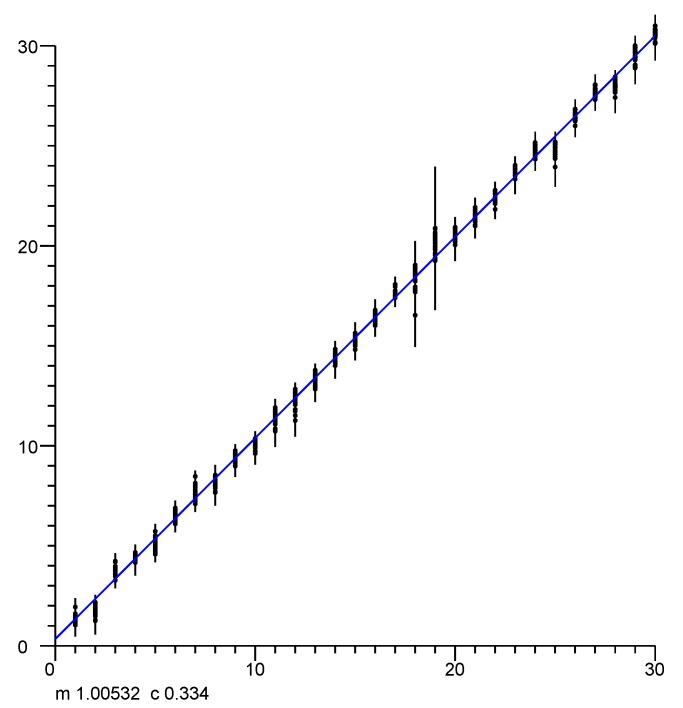
Linear fit of reported distance to actual distance (outdoors, clear line of sight (LOS), no obstructions in the first Fresnel zone). The offset in the well calibrated setup tested here happens to be small (less than half a meter), but can be five meters or more in other situations. Horizontal axis: actual position in meters. Vertical axis: fine time measurement (FTM) of the round trip time (RTT) reported distances in meters.

**Figure 5 sensors-20-01489-f005:**
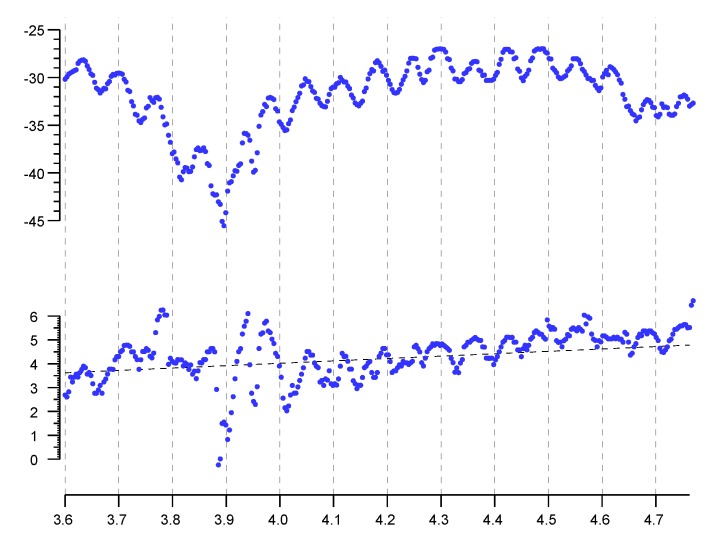
Upper plot: signal strength (RSS) in dBm, Lower plot: reported distance in meters. Horizontal axis: actual position in meters. Note undulations with wavelength somewhere between about half the wavelength and the full wavelength of the (2.4GHz) electromagnetic wave. (There is also an overall trend in RSS, probably due to destructive interference between the direct signal and a reflection off the concrete floor about 1.2 m below the line connecting the intiator to the responder).

**Figure 6 sensors-20-01489-f006:**
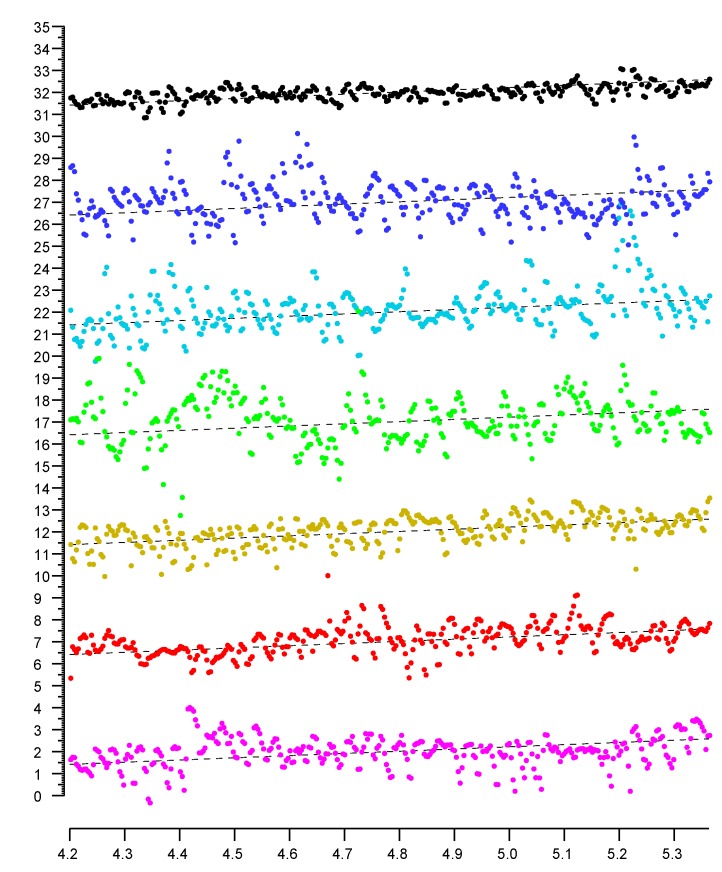
Sample plots (colored) of reported distances in six 80 MHz wide channels in the 5 GHz band. The top (black) plot is a simple average, which has less than half the error of the individual measurements. Horizontal axis: actual distance in meters. Vertical axis: reported distance in meters. (Note that the average slope of the plots is not equal to one, because the scales on the vertical axis and horizontal axes are different). (Individual plots are offset vertically to avoid overlap).

**Figure 7 sensors-20-01489-f007:**
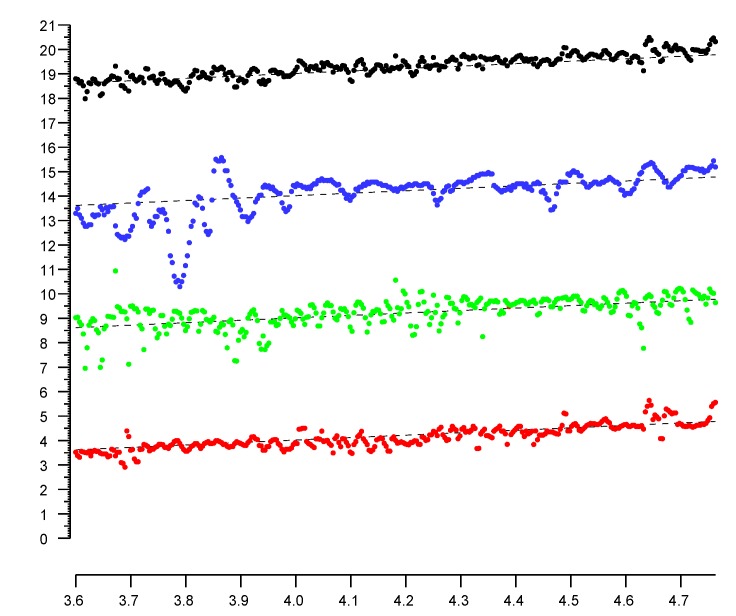
Sample plots (colored) of reported distances in three channels. The top (black) plot is a weighted average, which has only a bit over half of the average error in the individual plots. Horizontal axis: actual distance in meters. Vertical axis: reported distance in meters. (Note: the average slope of the plots is not equal to one, because the scales on the vertical axis and horizontal axes are different). (Individual plots are offset vertically to avoid overlap).

**Figure 8 sensors-20-01489-f008:**
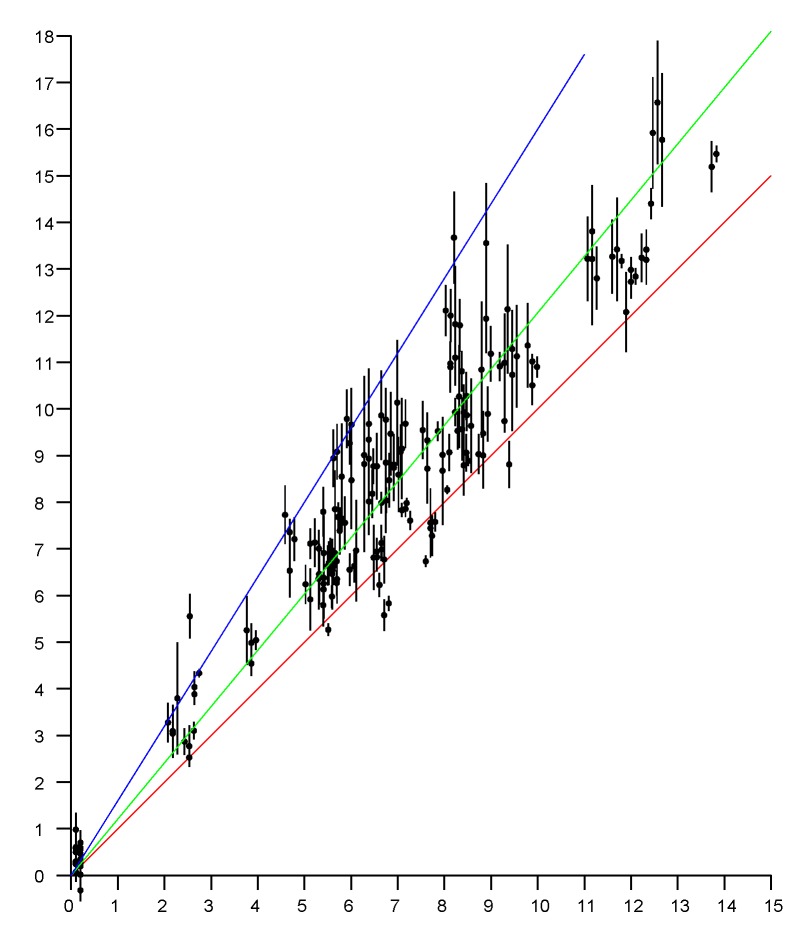
Scattergram of measured distances versus actual distance in wooden three-story house. Vertical axis: measured distance (meters). Horizontal axis: actual distance (meters). Red line (slope 1) is the ideal relationship; Green line (slope 1.2) is the best linear fit; Blue line (slope 1.6) is an upper extreme. The high permittivity of building materials biases the distances measured by FTM RTT.

**Figure 9 sensors-20-01489-f009:**
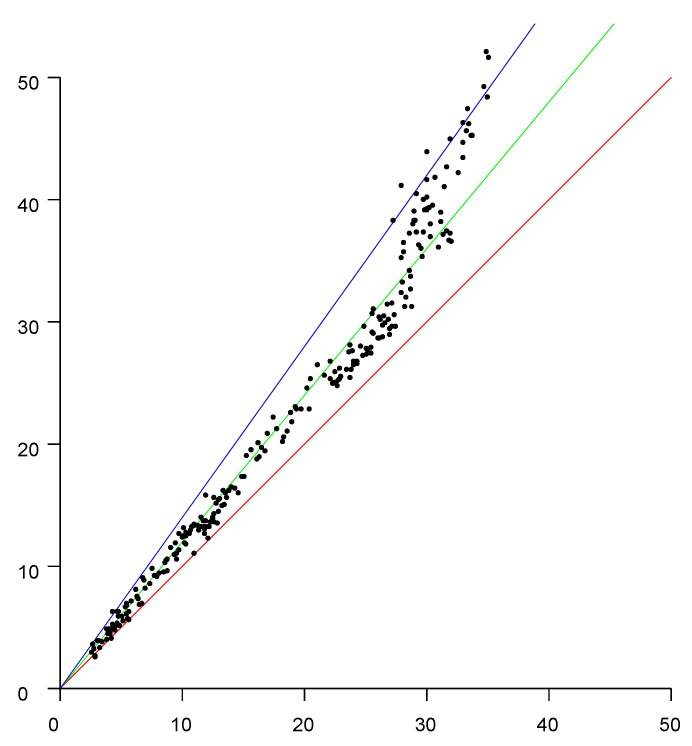
Scattergram of measured distances versus actual distance in a large open plan office building. Vertical axis: measured distance (meters). Horizontal axis: actual distance (meters). Red line (slope 1) is the ideal relationship. Green line (slope 1.2) is the best linear fit. Blue line (slope 1.4) is an upper extreme. The high permittivity of building materials biases the distances measured by FTM RTT.

**Figure 10 sensors-20-01489-f010:**
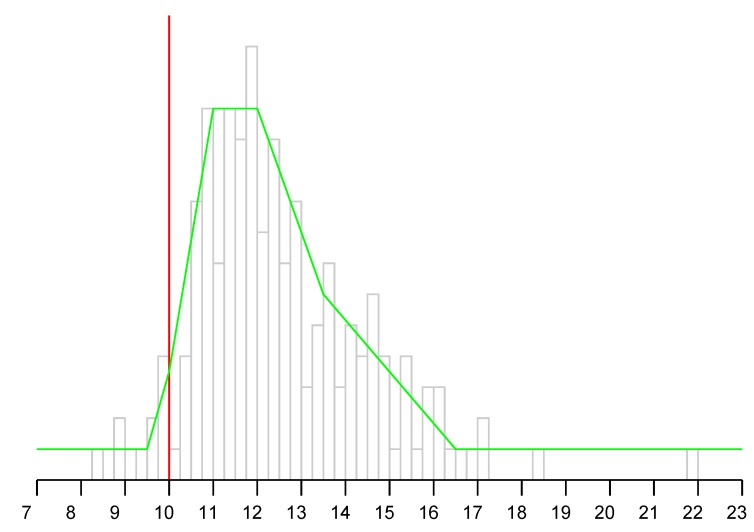
A slice through a particular observation model. Horizontal axis: measured distance (when actual distance is 10 m) Grey histogram: measurements from typical three-level residence. Green curve: observation model—probabilty of measuring the specified distance (piece-wise linear fit to grey histogram).

**Figure 11 sensors-20-01489-f011:**
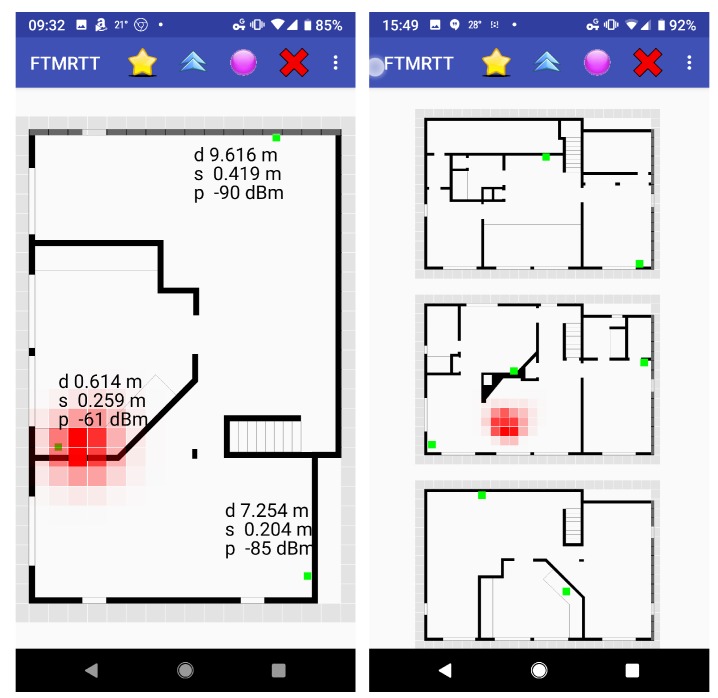
Sample “heat maps” of Bayesian grids. Left: 2-D case (single level) with 3 responders (green dots). Text shows current FTM RTT distance, st. dev. and signal strength. Right: 3-D case (three levels) with 7 responders. Voxels in each floor were collapsed into a single layer for display purposes.

**Figure 12 sensors-20-01489-f012:**
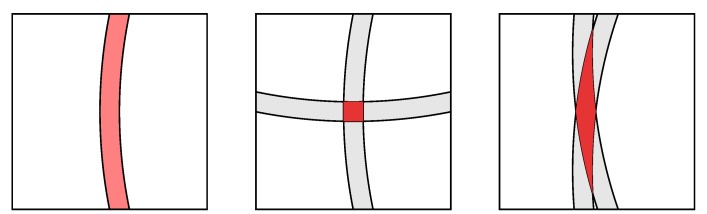
Dilution of Precision. Left: constraint from single distance measurement; Middle: favorable combination of constraints; Right: unfavorable combination of constraints. The area of the overlap grows as 1/sin(θ), where θ is the angle between the directions to the APs.

**Figure 13 sensors-20-01489-f013:**
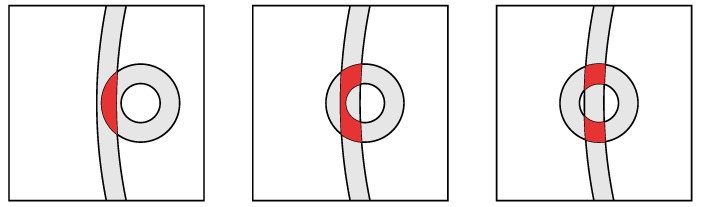
Dilution of Precision when close to responder. Left: Intersection is more or less an oblong oval; Middle: Intersection is approximately kidney shaped; Right: Intersection is bimodal. Such distributions cannot reasonably be approximated by multi-variate Gaussians.

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
