# Peer review of "Doubling the Accuracy of Indoor Positioning: Frequency Diversity"

_sensors, 2020, doi:10.3390/s20051489_

Round 1

Reviewer 1 Report

The paper presents an indoor localization methodology that is based on the fine time measurement introduced in the IEEE 802.11 rev. 2016. The author analyses the various sources of errors in WiFi-based indoor localization.

The paper is well written, well structured and easy to follow. Results are convincing and already appeared on the author's website in https://people.csail.mit.edu/bkph/

The author does not consider the effect of the AGC on RTT timing, which are reported in the following paper, which also analyses several sources of errors in measuring distances using time of flight.

Gallo P, Garlisi D, Giuliano F, Gringoli F, Tinnirello I. WMPS: A positioning system for localizing legacy 802.11 devices. Transactions on Smart Processing and Computing. 2012 Oct 2.

Fig. 2 of the paper referenced above provides a possible interpretation of some phenomena that also appear in Fig. 8 and 9 of the article under review and that are not explained.

Some artefacts that are reported in Fig. 2 and 3 are not adequately discussed (see attachment).

Fig. 4 reports red and black dots, what is the difference between them?

Fig. 5 presents a ripple, which is explained by the author, but there is also a trend (reported in the attachment in red), which is not explained. It would be interesting to interpret such results. 

The author by propose neglects the angle of arrival; we suggest the following reading 

Gallo P, Mangione S. RSS-eye: Human-assisted indoor localization without radio maps. In2015 IEEE International Conference on Communications (ICC) 2015 Jun 8 (pp. 1553-1558). IEEE.

We think that the empirical pdf reported in Fig. 10 depends on the position of the target, therefore cannot be used in a general algorithm. 

References [25-29] refer to Wikipedia; please, consider removing such references. 

Finally, the paper includes few typos and wrong punctuation, which are indicated in red in the attached file.

Author Response

The paper presents an indoor localization methodology that is based on the fine time measurement introduced in the IEEE 802.11 rev. 2016. The author analyses the various sources of errors in WiFi-based indoor localization.

The paper is well written, well structured and easy to follow. Results are convincing and already appeared on the author's website in https://people.csail.mit.edu/bkph/

I want to thank the reviewer for a very thorough analysis and very helpful suggestions and references!

The author does not consider the effect of the AGC on RTT timing, which are reported in the following paper, which also analyses several sources of errors in measuring distances using time of flight.
Gallo P, Garlisi D, Giuliano F, Gringoli F, Tinnirello I. WMPS: A positioning system for localizing legacy 802.11 devices. Transactions on Smart Processing and Computing. 2012 Oct 2.

Thank you very much for this reference, which I had not come across before. While I didn't claim in the paper to understand where the "position dependent" error actually comes from, and while the solution does not depend on understanding this, it is very helpful to have suggestions for the origin of this large error. I did venture to suggest a possible mechanism having to do with an adjustment (perhaps of some threshold) based on signal strength. Changing automatic gain control is related since the AGC will vary with signal strength. In fact, as far as I know, the RSSI reported is typically based on the AGC setting. It is very good to now have a reference that says more on the "suspected main cause"!

I've also removed the "for the first time" from the abstract.

Fig. 2 of the paper referenced above provides a possible interpretation of some phenomena that also appear in Fig. 8 and 9 of the article under review and that are not explained.

If I understand this correctly, Fig. 2 in the reference suggests that the reported distance can jump between discrete values when the AGC value changes. That is, discrete jumps in AGC will produce "jumps" in the error (although this seems a little different since it involves jumps of 60 meter in he reference). I can see that the red line added by the reviewer in Fig. 8 suggests similar behavior. But I wouldn't want to stick my neck out to claim for sure that this is the explanation. For one thing, the measurements in Fig. 8 are not sequentially obtained going left to right (where AGC may jump at discrete locations and then stay at some fixed value for some distance) but represent a "scattergram" of results from measurements obtained on a set of N APs in fixed location (By moving the initiator to each of the N locations in turn, a set of N*(N-1) measurements are obtained). The same applies to Figure 9, where again, the measurements are not obtained sequentially with increasing distance. Of course, it may be that the gain setting for a small range of distances is the same in some cases as long as the attenuation along the path is also about the same.

Some artifacts that are reported in Fig. 2 and 3 are not adequately discussed (see attachment).

There is a reason: I don't know where these small "glitches" come from. Which is too bad, but I think that does take away from the main point that overall repeated measurements in one location are remarkable constant and that this is totally misleading when estimating the "error". Perhaps equally importantly, they point out the "non-standard" (e.g. not at all i.i.d. Gaussian) nature of even this small "measurement noise".

Specifically in Fig 2. First, yes the 2.4 GHz measurement has much more "noise" when measured in the same location (keeping in mind that this measurement "noise" is small relative to the "position dependent" noise and hence not really important). Then, the area on the dark green plot outlined by the reviewer with a red circle illustrates a not uncommon phenomenan where there is more than one mode with the device switching back and forth between them. One guess about this is that in the super-resolution algorithm (as mentioned in the paper), the initial computation may pick the wrong interval in which to "super resolve". But, again I have no way to prove this. Fortunately it doesn't effect the conclusions of the paper.

Next, at the place marked by the red arrow on the grey plot, there appears to be a transition between two states. Again, I don't really know why this happens, but may conjecture again that it is some thing to do with the super resolution algorithm. Or perhaps AGC gain switching as suggested by the reviewer (but why would it suddenly switch and stay changed?). Again, I wish I understood these "glitches", but not understanding them does not stop one from agreeing with the main points of the paper, such as that the i.i.d Gaussian model of noise is not appropriate and that the "measurement noise" illustrated in Fig. 2 is actually much smaller than the "position dependent" noise and hence not of much interest.

In Figure 3, the reviewer also marked a couple of points. Again, I do not know why the dark green plot showed a large range of errors in the area marked and not in other areas. Maybe the signal strength got much lower. Then, marked with a red arrow is a temporary drop out of one of the responders. This would happen at times in experiments without obvious rhyme or reason. By the way, I also have no idea why some responders do better than others, but do know that one should not draw conclusions about the quality of an AP for FTM RTT purposes from just one such plot, because the rank ordering can be quite different in a different environment (which is also why I resisted naming the ten responders in question).

Fig. 4 reports red and black dots, what is the difference between them?

The black dots and black bars are the distance and st. dev. as reported by FTM RTT API (which on Android means averages over 7 trials). The red dots and red bars are means and averages over 16 such API calls. Rather than explain this side detail, I have redone the figure to show only one of them.

Fig. 5 presents a ripple, which is explained by the author, but there is also a trend (reported in the attachment in red), which is not explained. It would be interesting to interpret such results.

I believe this is due to a reflection off the concrete floor approximately 1.1 meter below the AP and the initiator caused causes destructive interference at a certain distance (path length difference is approximately a wavelength and reflection flips the polarity). But I thought it a distraction to explain this. I added a parenthetical comment now. To me what is important is that the figure supports the argument in the text.

The author by propose neglects the angle of arrival; we suggest the following reading

Gallo P, Mangione S. RSS-eye: Human-assisted indoor localization without radio maps. In2015 IEEE International Conference on Communications (ICC) 2015 Jun 8 (pp. 1553-1558). IEEE.

Thank you for the reference. I tried to avoid getting too deep into the ongoing "platform wars" between Bluetooth angle of arrival camp (Apple) and the WiFi FTM RTT (Android) camp. But have added a few more words and the reference to this. (Also, there is a paragraph already about AoA).

We think that the empirical pdf reported in Fig. 10 depends on the position of the target, therefore cannot be used in a general algorithm.

Yes. Agreed. The idea was that the observation model depends on the environment and so ideally would be constructed based on measurements. I've now made it clear that this is not meant to be a "general purpose" observation model that should be used in all cases (although, to be honest, in many demonstrations I have not bothered changing it despite being in different environments --- apparently it does not have to be very accurate to be useful).

References [25-29] refer to Wikipedia; please, consider removing such references.

OK, have done that.

Finally, the paper includes few typos and wrong punctuation, which are indicated in red in the attached file.

Thank you again; have taken care of them!

===============================================

There are also some handwritten comments/questions on a scanned page, that I thought I might benefit from, so I will respond to these comments here as well (I hope I transcribed them correctly).

"(2) Check the values as MP has a strong impact on the error"

I have not said a lot about multi-path, because, while it is widely blamed for the problems with FTM RTT, in my view it is not nearly as important as delay in materials of high relative permittivity. For a start, multi-path ideally should not affect the first arrival used in FTM RTT (ignoring second order effects discussed in the paper).

Then, when I turn my body so as to block the LOS, I believe I am seeing more the effect of the high relative permittivity along the direct path through the tissues than I am seeing things getting reflected off random furnishings around me (since this happens out doors away from reflectors as well). I don't have any good way of experimenting to prove this yet, but hope to make this the subject of a future paper.

"(3) is a well known result in the literature"

Yes, it is well known that RSSI is not good for indoor location, but many people don't know that and assume the inverse square law rules. The figure illustrates that actually exponential attenuation is a better model indoors. I have rephrased the text slightly and would be happy to refer to some standard reference here if the reviewer has a preferred reference.

"(4) Many WiFi cards actually provide CSI information, no need to have high-end measurement devices"

I do suggest using CSI instead of a network analyser, and indeed ever since 802.11n CSI has been available in the WiFi chip (e.g. Intel 5300). Sadly not so in Android API! I've been lobbying Google to make CSI available to apps (maybe this is easier on Unix, but that is not were the mass impact will be).

"(5) Why in Fig. 6 does not the ... 'texture' that is reported in Fig. 5? Both figures show 1 m range"

The measurements in Fig. 5 where selected to best show the phenomenon and used very fine sampling in the distance direction (That is, I picked the best plot of many to make the point). This fluctuation in reported distance is present in Fig. 6 as well, but not as prominent or as regular.

"(6) How are computed these weight?"

The weights were obtained by least squares fitting. They are the ones that produce the lowest error of the weighted average (although equal weight give only slightly worse results). Have added this to the text now.

"(7) What is the experimental result? Is it used a track to move the STA? What is the indoor environment?

Yes, the STA is moved in 3mm increments on a suspended styrofoam track (to minimize interference with the RF signal). The environment is a typical three level wooden structure with concrete basement walls. Have added some words to this effect to the text (was reluctant to give too many detail that might only lead to more questions).

"(8) How to define the weights?"

The weights were obtained by least squares fitting. They are the ones that produce the lowest error of the weighted average. Have added this to the text now.

"(9) The author assumes that the AP, STA are in LOS. Why?

I did not want to get into this, because there are such strong opinions about multi-path being important, but in my experience reflected signals are often too weak to be of importance (unless there is a really strong reflector such as metal sheet or a very low angle of reflection). Also, if the LOS is attenuated rather than completely blocked (e.g. by a metallic wall) then first arrival as measured by FTM RTT should not be affected by reflections. Hopefully future work will help clarify this. Perhaps more importantly, the observation model will be biased towards large distances in either case, whether the distance is overestimated because of a longer path or because of a path with higher relative permeability.

Again, thank you very much!

Reviewer 2 Report

The authors have listed and described many causes that contribute to RTT errors. The paper is well arranged and most of the causes are well explained, although some of them can be further improved. 

If the authors can provide potential solutions to dealing with such causes, the paper will be more valuable. 

Author Response

The authors have listed and described many causes that contribute to RTT errors.
The paper is well arranged and most of the causes are well explained, although some of them can be further improved.

Thank you. I hope that some of the ideas in the paper will help with this. In particular, the dominance of the "position dependent" error, which is here first described. It is quite possible that there are still things to be discovered about the causes of the errors. Hopefully future research will clarify this.

If the authors can provide potential solutions to dealing with such causes, the paper will be more valuable.

Well, there are some suggestions for dealing with the error causes in the paper. The main one, of course, is the user of frequency diversity to dramatically cut the main error, the "position-dependent" error. This is now implementable. Other solutions, such as using channel state information (CSI) has to wait for system developers to provide the proper API (Google in this case). At this point we have no ideas other than those already in the paper.

Reviewer 3 Report

The paper is quite interesting and deserves publication.

I suggest to add an explanation about why, in the Figs. 3, 6 and 7, the general trends show the same positive slopes : it took me a few minutes to understand that this slope is just the difference of scales in x and y.

Author Response

The paper is quite interesting and deserves publication.

Thank you.

I suggest to add an explanation about why, in the Figs. 3, 6 and 7, the general trends show the same positive slopes : it took me a few minutes to understand that this slope is just the difference of scales in x and y.

Thank you for the suggestion. Have added an explanation to the captions.

Reviewer 4 Report

Very interesting paper.

Like others, this paper uses the term ‘localisation’ to denote ‘position’. This is common use, but it might be better to stick to the term ‘position’ as the paper is about the accuracy and error in point positioning without taken the (indoor) geographical context into account, and not about localisation which takes into account the (indoor) geographical context. This reviewer (not being the author!) likes the definition given in Groves, 2013, Principles of GNSS, chapter 1.1:  “The terms position and location are nominally interchangeable, but are normally used to denote two different concepts. Thus, position is expressed quantitatively as a set of numerical coordinates, whereas location is  expressed qualitatively, such as a city, street, building, or room. A navigation system will calculate a position, whereas a person, signpost, or address will describe a location. A map or geographic information system (GIS) matches locations to positions, so it is a useful tool for converting between the two. Figure 1.2 illustrates this. Some authors use the term localization instead of positioning, particularly for short-range applications. The two are  essentially interchangeable, although “localization” is also used to describe techniques that constrain the position solution to a particular area, such as a street or room, instead of determining coordinates.”

If the author of this paper agrees with point (sec) of view, then the title should read: “Doubling the Accuracy of Indoor Positioning: Frequency Diversity”, and all occurences of “indoor location” would be altered in “indoor position”, and all occurances “fixed location” to be altered in “fixed position”.

This reviewer checked: https://people.csail.mit.edu/bkph/ftmrtt and here the author of this paper uses “Indoor positioning”. So, it might be the case the author of the paper agrees with this statement.

Some other remarks:

Abstract: “First responder” in the meaning of: 'the officer who is first on the location'. In this paper the term 'responder' is used for a Wi-Fi access point. Would be better to use another term for 'first responder' to avoid confusion.

III.A. Background: “to measure the signal strength”, abbreviation RSS, not RSSI (received signal strength index).

III.B. Fingerprinting: Would be nice to stress Fingerprinting, contrary to 'simple' Single Strength does not require the position of the multiple AP (responders),

IV.D. Noise Gain:  “referred to as “dilution of precision” (DOP) in GPS terminology.” Would be nice to include a reference in which DOP in the context of GPS is defined. See, again, Groves, 2013, Principles of GNSS, section 7.4.5. Effect of Signal Geometry. Compare also fig. 12 and fig. 13. in this paper with figure 7.21 in Groves, 2013.

Appendix D: Placement of responders

Please check: https://repository.tudelft.nl/islandora/object/uuid%3Ab91e1ec5-7c81-4fcc-b34c-654b9da2140c?collection=education

Author Response

Like others, this paper uses the term ‘localisation’ to denote ‘position’. This is common use, but it might be better to stick to the term ‘position’ as the paper is about the accuracy and error in point positioning without taken the (indoor) geographical context into account, and not about localisation which takes into account the (indoor) geographical context. This reviewer (not being the author!) likes the definition given in Groves, 2013, Principles of GNSS, chapter 1.1:

“The terms position and location are nominally interchangeable, but are normally used to denote two different concepts. Thus, position is expressed quantitatively as a set of numerical coordinates, whereas location is expressed qualitatively, such as a city, street, building, or room. A navigation system will calculate a position, whereas a person, signpost, or address will describe a location. A map or geographic information system (GIS) matches locations to positions, so it is a useful tool for converting between the two. Figure 1.2 illustrates this. Some authors use the term localization instead of positioning, particularly for short-range applications. The two are essentially interchangeable, although “localization” is also used to describe techniques that constrain the position solution to a particular area, such as a street or room, instead of determining coordinates.”

If the author of this paper agrees with point (sec) of view, then the title should read: “Doubling the Accuracy of Indoor Positioning: Frequency Diversity”, and all occurrences of “indoor location” would be altered in “indoor position”, and all occurrences “fixed location” to be altered in “fixed position”.

This reviewer checked: https://people.csail.mit.edu/bkph/ftmrtt and here the author of this paper uses “Indoor positioning”. So, it might be the case the author of the paper agrees with this statement.

OK, very interesting. Blame it on ESL! I have made the changes, which includes a change in the title --- which I will have to ask the editor to adjudicate.

Interestingly the IEEE 80211-2016 standard refers to *both* of the above-mentioned concepts as types of "locations": Location Configuration Information (LCI) can provide latitude, longitude, altitude and their uncertainties. Location Civic Report (LCR or CIVIC) can provide a ``civic'' address in a standardized key-value format (such as street address, city etc.)

Also, in machine vision and robotics the term "localization" is used for positioning --- for example, SLAM is an acronym for "Simultaneous Location and Mapping" a method that builds a map of the environment while also determining where the robot is "positioned" :-)
https://en.wikipedia.org/wiki/Simultaneous_localization_and_mapping

Also, in Google Android API, reference is to "location services". For example, one has to enable ACCESS_FINE_LOCATION, ACCESS_COARSE_LOCATION
https://developer.android.com/reference/android/Manifest.permission#ACCESS_COARSE_LOCATION
under "Settings > Location" and so on.
https://support.google.com/accounts/answer/3467281?hl=en
https://developer.android.com/training/location/change-location-settings

Also, the WiFi Alliance (which certifies various IEEE 802.11 WiFi capabilities)
certifies devices for "Wi-Fi location". See e.g.
Wi-Fi CERTIFIED Location™: Indoor location over Wi-Fi® (2017)
https://www.wi-fi.org/file/wi-fi-certified-location-indoor-location-over-wi-fi-2017

Anyway, despite all that, I took the reviewers recommendation and made the change throughout. The editor may have the final word on this. I'm happy either way.

Some other remarks:
------------------

Abstract: “First responder” in the meaning of: 'the officer who is first on the location'. In this paper the term 'responder' is used for a Wi-Fi access point. Would be better to use another term for 'first responder' to avoid confusion.

Good suggestion. Changed to "emergency worker"

III.A. Background: “to measure the signal strength”, abbreviation RSS, not RSSI (received signal strength index).

Thank you. I see: "dBm and RSSI are different units of measurement that both represent the same thing: signal strength. The difference is that RSSI is a relative index, while dBm is an absolute number representing power levels in mW (milliwatts)." Changed. 

III.B. Fingerprinting: Would be nice to stress Fingerprinting, contrary to 'simple' Single Strength does not require the position of the multiple AP (responders),

Ah. Excellent point. Added text.
"This method does not require knowledge of the positions of the APs"

IV.D. Noise Gain: “referred to as “dilution of precision” (DOP) in GPS terminology.” Would be nice to include a reference in which DOP in the context of GPS is defined. See, again, Groves, 2013, Principles of GNSS, section 7.4.5. Effect of Signal Geometry. Compare also fig. 12 and fig. 13. in this paper with figure 7.21 in Groves, 2013.

Have added reference to Groves 2013 reference work in two places, including Appendix "Placement of Responders".

Appendix D: Placement of responders

Please check: https://repository.tudelft.nl/islandora/object/uuid%3Ab91e1ec5-7c81-4fcc-b34c-654b9da2140c?collection=education
"A statistical analysis on the system performance of a Bluetooth Low Energy indoor positioning system in a 3D environment"

Have added reference to this also. Thank you for that reference!